# Profiling human breast epithelial cells using single cell RNA sequencing identifies cell diversity

Quy H. Nguyen[1], Nicholas Pervolarakis[2], Kerrigan Blake[2], Dennis Ma[1], Ryan Tevia Davis[3], Nathan James[1], Anh T. Phung[3], Elizabeth Willey[4], Raj Kumar[4], Eric Jabart[5], Ian Driver[4], Jason Rock[4], Andrei Goga [6], Seema A. Khan[7], Devon A. Lawson[3], Zena Werb [4] & Kai Kessenbrock[1]

Breast cancer arises from breast epithelial cells that acquire genetic alterations leading to subsequent loss of tissue homeostasis. Several distinct epithelial subpopulations have been proposed, but complete understanding of the spectrum of heterogeneity and differentiation hierarchy in the human breast remains elusive. Here, we use single-cell mRNA sequencing (scRNAseq) to profile the transcriptomes of 25,790 primary human breast epithelial cells isolated from reduction mammoplasties of seven individuals. Unbiased clustering analysis reveals the existence of three distinct epithelial cell populations, one basal and two luminal cell types, which we identify as secretory L1- and hormone-responsive L2-type cells. Pseudotemporal reconstruction of differentiation trajectories produces one continuous lineage hierarchy that closely connects the basal lineage to the two differentiated luminal branches. Our comprehensive cell atlas provides insights into the cellular blueprint of the human breast epithelium and will form the foundation to understand how the system goes awry during breast cancer.

[1] Department of Biological Chemistry, University of California, Irvine, Irvine, CA 92697, USA. [2] Center for Complex Biological Systems, University of California, Irvine, Irvine, CA 92697, USA. [3] Department of Physiology and Biophysics, University of California, Irvine, Irvine, CA 92697, USA. [4] Department of Anatomy and Biomedical Sciences Program, University of California, San Francisco, CA 94143-0452, USA. [5] ProteinSimple, 3001 Orchard Parkway, San Jose, CA 95134, USA. [6] Department of Cell and Tissue Biology, University of California, San Francisco, CA 94143-0452, USA. [7] Department of Surgery, Feinberg School of Medicine, Northwestern University, Chicago, IL 60611, USA. These authors contributed equally: Quy H. Nguyen, Nicholas Pervolarakis. Correspondence and requests for materials should be addressed to Z.W. (email: zena.werb@ucsf.edu) or to K.K. (email: kai.kessenbrock@uci.edu)

Breast cancer is a highly heterogeneous disease that is sub-typed based on tissue morphology and molecular signatures[1]. At least six different intrinsic subtypes of breast cancers have been established, namely luminal A, luminal B, HER2-enriched, basal-like, normal breast, claudin-low[2], and more recently up to ten subtypes have been described[3]. Each subtype is speculated to arise from a different cell of origin[4]; however, gaps in our understanding of the full spectrum of cellular heterogeneity and the distinct cell types that comprise the human breast epithelium hinder our ability to investigate their roles in cancer initiation and progression.

Breast cancer arises from the breast epithelium, which forms a ductal network embedded into an adipose tissue that connects the nipple through collecting ducts to an intricate system of 12–20 lobes, which are the milk producing structures during pregnancy and lactation. Throughout the duct and lobular system, the breast epithelium is composed of two known cell types, an inner layer of secretory luminal cells and an outer layer of basal/myoepithelial cells. A series of recent reports have indicated that further heterogeneity exists within these two cell layers in mice[4]. Two landmark papers published in 2006 identified a functionally distinct subpopulation of basal epithelial cells that harbors stem cell capacity and is capable of reconstituting a fully developed mammary epithelial network when transplanted into the cleared mammary fat pads of mice[5,6]. Moreover, a subpopulation of luminal progenitor cells identified by high expression of KIT as well as a subpopulation of mature luminal cells have been identified using flow cytometry (FACS) isolation strategies[7,8]. Interestingly, based on comparative bulk expression analyses, these luminal progenitors may have increased propensity to give rise to triple negative breast cancers in patients with mutations in the BRCA1 gene[9]. It remains to be determined if other distinct cell types exist within the breast epithelium and how these relate to the known subtypes of breast cancer.

Advances in next generation sequencing and microfluidic based handling of cells and reagents now enable us to explore cellular heterogeneity on a single cell level and reconstruct lineage hierarchies using single-cell mRNA sequencing (scRNAseq)[10,11]. This approach allows an unbiased analysis of the spectrum of heterogeneity within a population of cells, since it utilizes transcriptome reconstruction from individual cells. scRNAseq has been successfully applied to understand the complex subpopulations in normal tissues such as lung[11] or brain[10] as well as in various cancers including melanoma[12], glioblastoma[13], and within circulating tumor cells from patients with pancreatic cancer[14].

The goal of the present study is to generate a molecular census of cell types and states within the human breast epithelium using unbiased scRNAseq. Focusing on the breast epithelium, our work provides a critical first impetus toward generating large-scale single cell atlases of the tissues comprising the human body as part of the international human cell atlas initiative[15]. This molecular census can shed light on lineage relationships and differentiation trajectories in the human system and how it relates to breast cancer. Our single-cell transcriptome analysis provides unprecedented insights into the spectrum of cellular heterogeneity within the human breast epithelium under normal homeostasis and will serve as a valuable resource to understand how the system changes during early tumorigenesis and tumor progression.

## Results

### scRNAseq reveals three cell types in the breast epithelium.
We collected a cohort of reduction mammoplasties from age- and ethnicity-matched, post-pubertal and pre-menopausal females (Supplementary Data 1), and performed scRNAseq on purified breast epithelial cells, which were isolated from surrounding stromal cells using flow cytometry based on differential expression of CD49f and EpCAM[16]. Basal and luminal cells were separately loaded onto the Fluidigm C1 microfluidics-enabled scRNAseq platform (Fig. 1a). Capture efficiency was monitored by microscopic imaging to exclude doublets and debris from further analysis (Supplementary Fig. 1a, b). We used 13 C1 chips in total to capture and sequence transcriptomes of 868 cells from three human individuals. The resulting single cell cDNA libraries were sequenced in parallel at an average read depth of 1.6 M reads per cell. After removing cells with less than 900 genes detected and additional quality control filtering (see Methods section), we proceeded to analyze 703 single cell at ~4500 genes detected on average per cell, where the gene detection range was comparable between basal and luminal cells (Supplementary Fig. 1c).

To identify the main cell types within the breast epithelium that are generalizable across individuals, we performed a combined analysis of all cells from the three individuals using the recently described Seurat pipeline[17]. This analysis identified three very distinct clusters of cells (Fig. 1b), indicating that the breast epithelium is composed of three main cell types. We then explored the genes that are significantly up-regulated within each cluster (Fig. 1c), which revealed that these main clusters correspond to one major basal (KRT14+; AUC = 0.83) cell type and two luminal cell types that both express the typical markers KRT8 and KRT18. Importantly, cells representing all three cell types were detected in each of the three individuals (Supplementary Fig. 1d). We found several distinct markers for these luminal cell types such as SLPI (AUC = 0.89) for L1, and ANKRD30A (AUC = 0.81) for L2 (Supplementary Fig. 1e). Comparing these signatures to previously published microarray expression analyses of FACS-isolated human breast epithelial cells[9,18], we found that L1 corresponds closely to the CD49f+/EpCAM+ population designated as "luminal progenitors," L2 resembles the CD49f−/EpCAM+ population called "mature luminal," and the basal cluster matched with CD49f[hi]/EpCAM− "Basal/MaSC." Since basal cells contain a subset of mammary stem cells (MaSCs)[5,6,19], we examined the basal cell cluster in more detail. Particularly intriguing was the observation of a subset with increased expression of mesenchymal and stem cell markers ZEB1[20] and TCF4 (Fig. 1d). Interestingly, previous work established a direct link between mesenchymal gene expression signatures and MaSC capacity[21], suggesting these ZEB1/TCF4-expressing cells may represent a subset of basal cells with increased MaSC potential.

### Droplet-mediated scRNAseq reveals subpopulation diversity.
To determine whether additional cellular diversity exists, we next utilized a more scalable droplet-mediated scRNAseq platform (10× Genomics Chromium)[22]. Here, we focused on reduction mammoplasty samples from nulliparous women (Supplementary Data 1) to reduce variability associated with pregnancy-related changes of the breast. We isolated both luminal and basal cells together (EpCAM+/CD49f[hi/lo]) by flow cytometry and subjected them as one sample to droplet-based scRNAseq targeting on average 5000 cells per sample (Fig. 2a). We sequenced a total of 24,646 cells from four individuals (Ind4-7) at an average ~60,000 reads per cell.

After quality control filtering to remove cells with low gene detection (<500 genes) and high mitochondrial gene coverage (>10%), detailed clustering analysis of the first individual (Ind4) using Seurat confirmed the existence of three main epithelial cell types, namely Basal (KRT14+), Luminal1 (L1; KRT18+/SLPI+) and Luminal2 (L2; KRT18+/ANKRD30A+) (Fig. 2b). All marker genes are listed in Supplementary Data 2. These analyses also

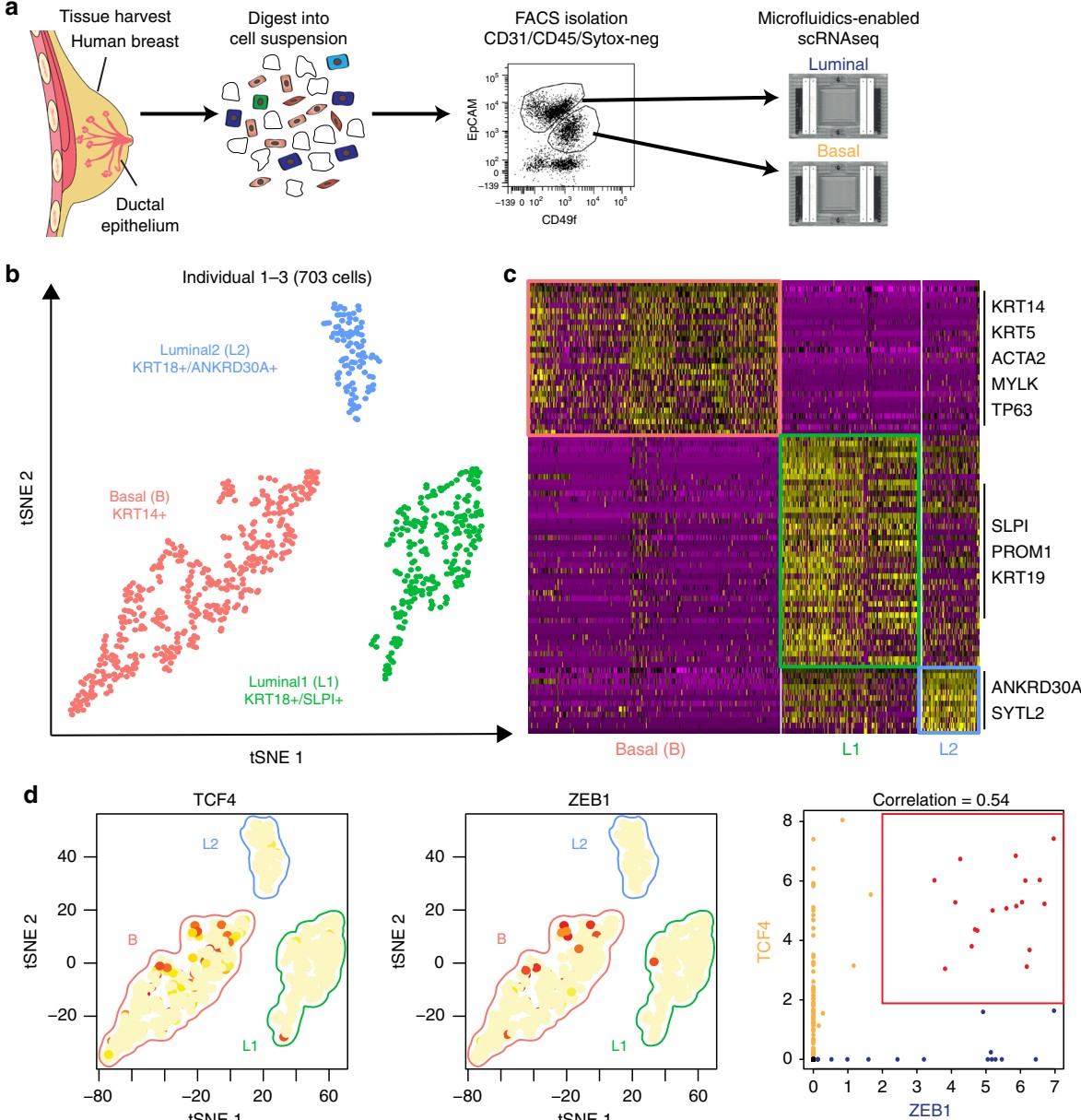

**Fig. 1** Identification of three major epithelial cell types and their markers using scRNAseq. **a** Overview of scRNAseq approach using primary human breast tissue samples that were processed into single cell suspension, followed by FACS isolation of basal (CD49f-hi, EPCAM+) and luminal (CD49f+, EPCAM-hi), and scRNAseq analysis using the microfluidics-enabled scRNAseq. **b** Combined tSNE projection of cells from all three microfluidics-enabled scRNAseq datasets. The major basal cluster is highlighted in red; Luminal1 (L1) in green; Luminal2 (L2) in blue. **c** Heatmap displaying the scaled expression patterns of top marker genes within each cell type with selected marker genes highlighted; yellow indicating high expression of a particular gene, and purple indicating low expression. **d** Feature plots showing the scaled expression of TCF4 and ZEB1 marking a subpopulation of basal cells and gene plot showing co-expression of TCF4 and ZEB1 in the same cells. See Supplementary Fig. 1 capture site imaging, gene detection, individual principal component analysis, tSNE plot colored by individual-derived cells and feature plots of cell type-specific markers

revealed three additional small clusters; cluster 8 was defined by stromal marker *VIM* ($P < 9.6 \times 10^{-25}$); cluster 9 showed specific expression of endothelial marker gene *ESAM* ($P < 4.1 \times 10^{-30}$); and cluster 10 included a small number of dispersed cells most likely representing outliers. We concluded that these clusters (8–10) were of non-epithelial nature and denoted them as unclassified (X) in further analyses.

Interestingly, multiple subclusters emerged within each of the main epithelial cell types as indicated by their distinct marker gene signatures (Fig. 2c). We hypothesized that the main islands of cells (Basal, L1, L2) represent distinct "cell types", whereas subclusters within each island depict "cell states" that are more

transient over time[23]. Within basal cells we detected three distinct cell states, which showed specific expression of inflammatory mediators (*IL24*; $P < 1.4 \times 10^{-180}$; Cluster 3), markers for myoepithelial cell function (*ACTA2*; $P < 7.4 \times 10^{-292}$; Cluster 4) and specific epithelial keratin expression (*KRT17*; $P < 1.6 \times 10^{-38}$; Cluster 5), respectively. *ZEB1* and *TCF4*, which marked a subset of basal cells in our microfluidics-enabled scRNAseq analysis (Fig. 1d), were lowly detected and therefore not interpretable in droplet-enabled scRNAseq, which is likely due to lower coverage compared to the microfluidics-enabled platform[24].

Within luminal cell type L1 we observed three distinct cell states that were marked by genes associated with milk production

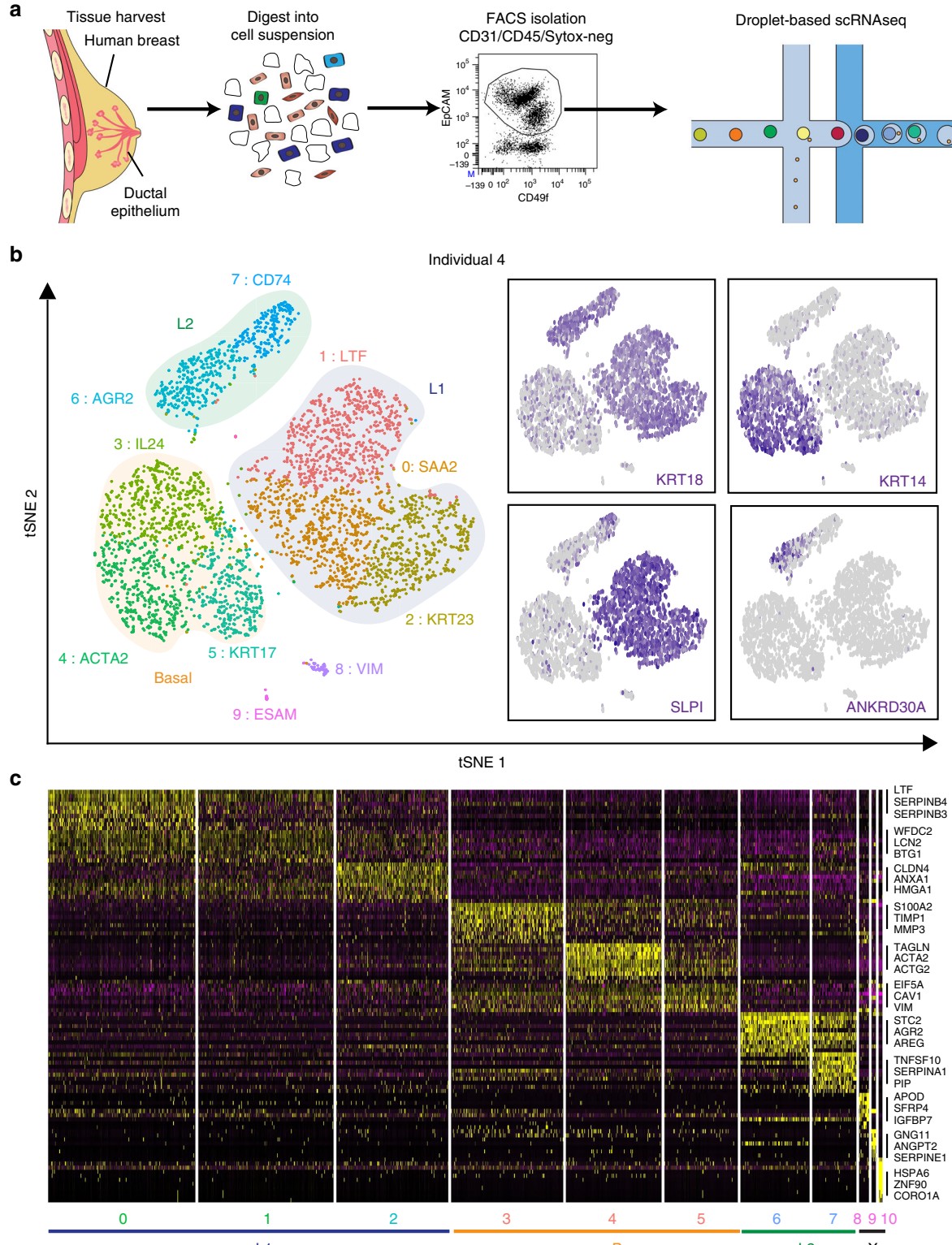

**Fig. 2** High throughput droplet-mediated scRNAseq reveals additional epithelial cell states. **a** Overview for droplet-enabled scRNAseq approach as described above; basal and luminal epithelial cells were sorted together and subjected to combined scRNAseq analysis using the droplet-based scRNAseq. **b** Data from individual four was analyzed using Seurat and the distinct clusters (0–10) are displayed in tSNE projection with selected marker gene for each cluster, and main epithelial cell types (Basal, L1, L2) are outlined. Feature plots of characteristic markers for the three main cell types are shown on the right showing expression levels as gradient of purple. **c** Heatmap showing the top ten marker genes for each cluster as determined by Seurat analysis with three selected genes per cluster highlighted on the right. See Supplementary Fig. 2 for individual clustering and marker gene analyses for Individuals 5–7

($LTF$; $P < 8.4 \times 10^{-270}$; Cluster 1), high expression of secretory molecules ($SAA2$; $P < 2.2 \times 10^{-90}$; Cluster 0) and distinct epithelial keratin expression ($KRT23$; $P < 2.5 \times 10^{-157}$; Cluster 2). The second luminal cell type L2 harbored two distinct cell states that were marked by expression of hormone responsive genes ($AGR2$; $P < 3.1 \times 10^{-144}$; Cluster 6) and specific cell surface markers ($CD74$; $P < 2.9 \times 10^{-121}$; Cluster 7). We next performed detailed individual Seurat clustering analyses for three additional individual datasets from nulliparous women, which confirmed many of the patterns described for Ind4 (Fig. 2). Like Ind4, the other individuals possessed three main cell clusters clearly corresponding to cell types Basal, L1, and L2, and eight to ten subclusters (Supplementary Fig. 2a–c). The number of subclusters per cell type varied across the individuals with Ind5 comprising five Basal, three L1 and one L2 clusters, Ind6 containing seven Basal, three L1 and one L2 clusters, and Ind7 comprising one Basal, three L1 and five L2 clusters (Supplementary Fig. 2a–c), which may be due to individual-to-individual variation or anatomical location of the surgical specimens.

To determine cell states that are generalizable across individuals, we developed a comparative approach using a cell scoring method adapted from recently published work[12]. Using the marker gene signatures for each of the 11 clusters (0–10) detected in Ind4 (Fig. 2b, c), we performed pairwise gene scoring analyses to find matches for every distinct cluster identified in Ind5–7 (Supplementary Fig. 2a–c). Comparing Ind4 to Ind5–7 showed that the main cell types (Basal, L1, L2) readily match up across all individuals (Fig. 3a–c). In addition, it revealed that the there are two distinct cell states present within L1 (L1.1 and L1.2) that emerge in all four individuals. The L2 population, which contained two clusters in Ind4, was found to be more homogeneous, and therefore these clusters were combined to a single L2 population. Comparing basal subclusters between individuals suggested that there are at least two generalizable cell states within basal cells (Fig. 3a–c). To further explore this, we performed a separate Seurat analysis using combined basal cells from all four individuals (Supplementary Fig. 3a). Several clusters displayed consistently high expression of genes associated with myoepithelial cell function (e.g., $ACTA2$, $TGLN$, $KRT14$). We therefore generated a "myoepithelial cell signature" gene list (Supplementary Data 2) based on published work[25] to stratify basal cells into either a "Basal" or "Myoepithelial" grouping

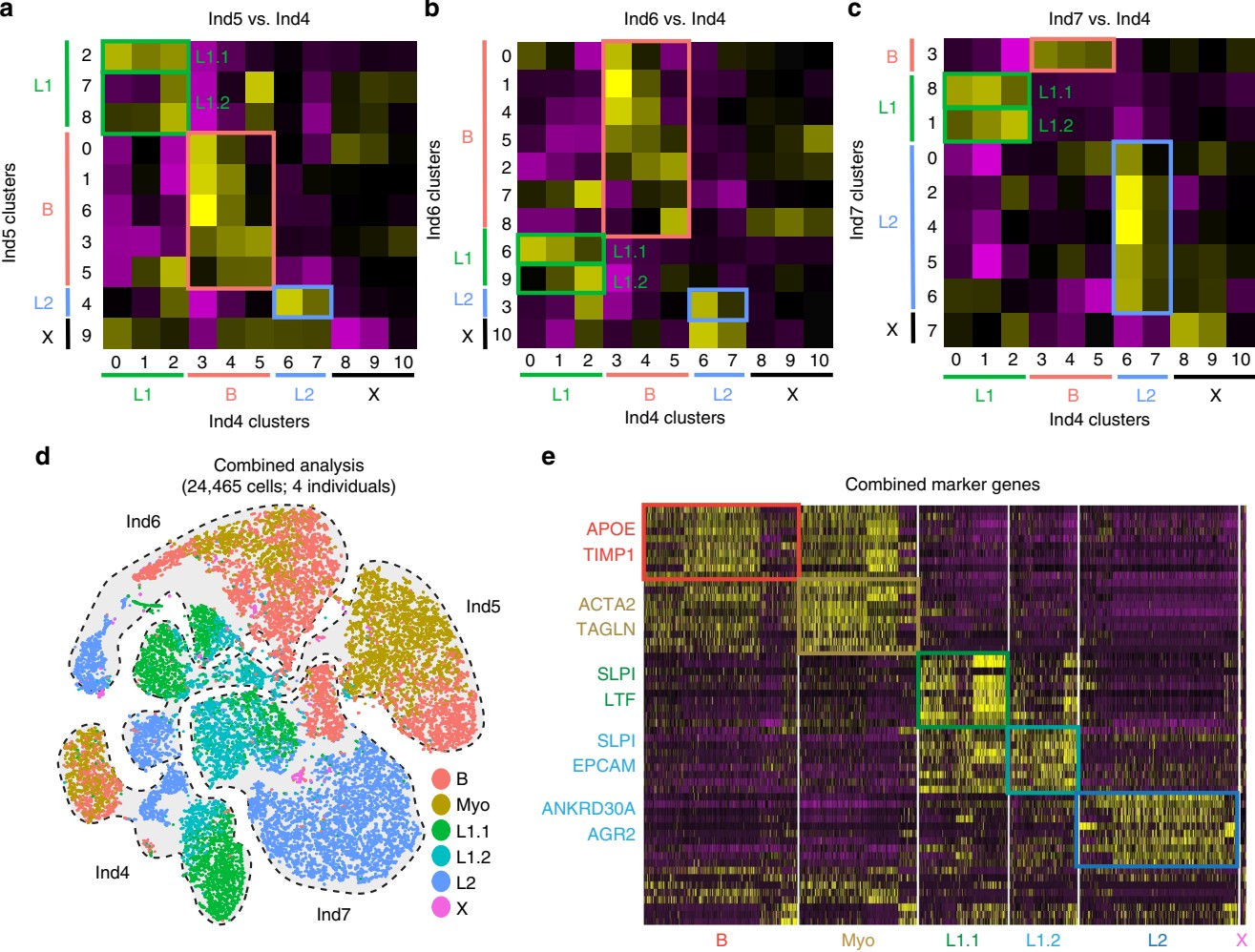

**Fig. 3** Combined droplet based RNAseq data to identify generalizable cell types and states. **a–c** Heatmaps showing gene scoring results using marker genes for Ind4 clusters (0–10; on bottom of heatmap) in all clusters from Ind5 (**a**), Ind6 (**b**), and Ind7 (**c**). Individual-specific cluster IDs are shown in different colors on the right and bottom, and cell type IDs for Basal (**b**), L1, L2, X are indicated on for every cluster. Data shown as $Z$ scores from purple (low) to yellow (high). Two distinct cell states L1.1 and L1.2 were found within L1 in all pairwise comparisons as highlighted by colored boxes on heatmap. **d** Combined tSNE projection of all individual datasets (outlined) is shown including the cell state identity marked by different colors. **e** Heatmap showing the expression pattern of the top ten markers per cell state with selected markers indicated (yellow = high expression; purple = low expression). See Supplementary Fig. 4 for separate basal cell Seurat analysis, summary of cell state designations and Ingenuity Pathway Analysis

(Supplementary Fig. 3b, c). These results allowed us to include all individual-specific clusters into the final cluster designations, namely Basal (B), Myoepithelial (Myo), Luminal1.1 (L1.1), Luminal1.2 (L1.2), Luminal2 (L2), and the small Unclassified (X) as summarized in Supplementary Fig. 3c. These designations were used to perform a combined Seurat analysis of all 24,465 cells from four individuals (Fig. 3d), which enabled us to determine the common marker genes (e.g., B: *APOD*; Myo: *TAGLN*; L1.1: *LTF*; L1.2: *CLDN4*; L2: *AGR2*) for each cell state that are generalizable across all four individuals (Fig. 3e).

To learn more about the biology underlying these cell states, we used Ingenuity Pathway Analysis (IPA) to identify distinct signaling pathways (Supplementary Fig. 3d), and interrogated for transcription factor consensus sites using the Enrichr tool[26] (Supplementary Data 2). These analyses revealed that the Myo state might be controlled by the transcription factors TP63 and PPARγ, and is defined by increased integrin and paxillin signaling indicating that these cells provide physical integrity within the breast epithelial architecture. The B state was found to be linked to transcription factors STAT3 as well as SOX2, NANOG, and KLF4, which are associated with stem cell capacity and cellular plasticity[27], suggesting that population B may harbor MaSCs. Within the luminal compartment, L1.1 showed distinct signatures of iNOS and IL6 signaling that may indicate a sentinel function of tissue harm and inflammation associated with this cell state. L1.2 displayed increased levels of *PI3K/AKT* and glucocorticoid signaling, which may indicate a link to steroid hormone signaling for this cell population. Within the second luminal cell type L2 we found evidence for elevated mTOR signaling as well as aldosterone signaling in epithelial cells, which suggests that this cell type represents a hormone-responsive cell population.

**Spatial integration of cell types and states**. We next used indirect immunofluorescence analysis to validate our scRNAseq findings on the protein level and to spatially integrate newly discovered cell types and states into the anatomy of the breast. We first focused on the cell states detected within the basal compartment. Immunostaining for ZEB1, which we identified in a subset of basal cells in microfluidics-enabled scRNAseq (Fig. 1d), showed that this protein is indeed expressed in a small fraction of basal epithelial cells (Fig. 4a). High ZEB1 and medium KRT14 levels have been recently described in a population of protein C receptor (ProCR) expressing murine MaSCs with in vitro and in vivo stem cell activity[19]. Comparison of published gene expression signatures of ProtCR+ MaSCs with the ZEB1+ population identified here showed striking similarity (Fig. 4b), suggesting that the ZEB1+ basal cells may represent a population of human MaSCs. In addition, staining for TCF4, revealed a comparable staining pattern to ZEB1 within the basal (smooth muscle actin-positive) compartment (Fig. 4c). These findings show that the cell state characterized by ZEB1 and TCF4 expression exists within the basal compartment in intact breast tissue.

KRT14 expression is a hallmark for basal cells, and our differential gene expression analysis confirmed that KRT14 is predominantly expressed within basal cells. However, it exhibited surprising variability across all basal cell population with particularly high expression in the Myo cell state (Fig. 4d). Immunofluorescence analysis for KRT14 confirmed this, and revealed that KRT14 high cells localized to the basal cell layer within ductal regions, while lobular basal cells generally displayed lower and more variable staining for KRT14 (Fig. 4e). Myo cells also expressed high levels of the definitive myoepithelial marker *ACTA2* (Supplementary Fig. 4a), as well as other genes associated

with smooth muscle differentiation and function in other tissues such as *MYLK*, *MYL9*, and *TAGLN/Transgelin*[28].

Surprisingly, basal and luminal markers were not always exclusive and we noted a distinct fraction of cells that co-express luminal- (e.g., *KRT8*) and basal- (e.g., *KRT14*) specific genes, as shown by correlation analysis of our single cell expression data (Supplementary Fig. 4b). To determine whether this population exists in the intact tissue, we performed in situ co-localization analysis by immunofluorescence staining for KRT8 and KRT14. While most areas within the human breast epithelium showed the expected luminal KRT8+/KRT14− or basal KRT8−/KRT14+ pattern, we observed several rare loci within lobular regions of the tissue that indeed showed distinct KRT8+/KRT14+ patterns (Supplementary Fig. 4c). Although this cell state has been previously observed in mouse fetal MaSCs[29], our work revealed that this state exists in the human tissue in adult homeostasis.

The scRNAseq analyses revealed that the luminal compartment harbors two discrete epithelial cell types (L1, L2). To determine if L1 and L2 correspond to ductal and lobular anatomical location within the tissue, we used specific markers for L1 (SLPI) and L2 (ANKRD30A) to identify their spatial distribution within the breast tissue using in situ immunofluorescence. These analyses showed that both L1 and L2 are located next to each other within both ducts and lobules (Fig. 5a). We next sought to determine their hormone signaling status. Annotation of the single cell datasets shows that L2 is particularly enriched for *ESR1*, *PGR*, and *AR* (Supplementary Fig. 5a), although generally these genes were found to be lowly expressed. Consistent with this observation, we also found on the protein-level that L2 marker ANKRD30A commonly overlaps with ER (32.4% of cells), PR (38.0%), and AR (46.8%), whereas SLPI-positive cells showed markedly lower percentage of hormone receptor expression (Fig. 5b–d). *PGR* was also expressed in a sub-fraction of basal cell states, although PR was not detected in basal cells on the protein level (Fig. 5c).

Proliferation is associated with active progenitor cell capacity within adult epithelial tissues[30]. Interestingly, we observed proliferative cells in all three epithelial cell types (Basal, L1, L2) as evident on the RNA level (Supplementary Fig. 5b) and using Ki67 immunostaining (Fig. 5e–f). Moreover, expression of *CDKN1B* (p27), which has been previously linked with a quiescent, hormone-responsive progenitor cell population[31], was found highest in L2 (Supplementary Fig. 5b), while markers for alveolar luminal progenitor cell function such as ELF5[32] and KIT[7,8] were specifically enriched in luminal subpopulation L1.1 (Supplementary Fig. 5c).

L2 was also characterized by higher levels of KRT8 than L1 (Fig. 5g). To quantify protein expression in individual cells, we utilized a recently developed single-cell western blot application (ProteinSimple, Milo), which performs electrophoretic separation of the protein content of about 2000 cells per chip and subsequently probed with fluorescently labeled antibodies. Applying single-cell western blotting to luminal and basal cells isolated by FACS identified three cell states, namely KRT8-negative, -low, and -high (Fig. 5h–i), which illustrates the usefulness of single cell Western blotting as a quantitative validation tool downstream of scRNAseq analyses.

Taken together, these analyses confirmed remarkable concordance between the patterns observed in scRNAseq and on the protein-level in intact tissues. Our spatial analyses confirmed that the luminal compartment contains two distinct cell types (L1 and L2) that intermingle within ducts and lobules. Both contain a subset of proliferative cells, suggesting that they each contain L1- and L2-committed progenitor cells to maintain these cell types. Based on their expression signatures, L1 may be committed to secretory function, while L2 likely functions as a hormone-sensing unit of the breast epithelium.

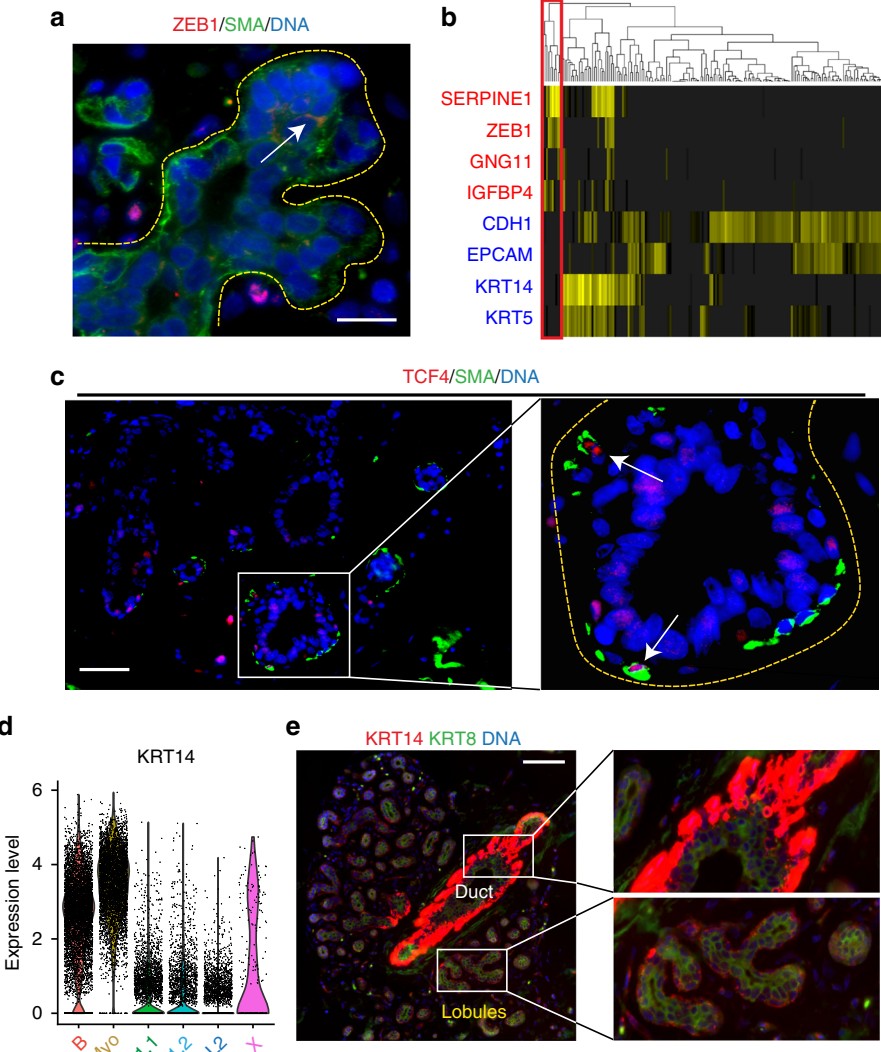

**Fig. 4** Characterization and spatial integration of basal cell states. **a** Immunofluorescence analysis of ZEB1 protein expression (red) in combination with basal marker KRT14 (green) and DNA stain using DAPI (blue) within tissue sections from primary human reduction mammoplasty samples showing ZEB1 expression in a subpopulation of basal (KRT14+) cells. Scale bar = 15 μm. **b** Heatmap showing expression of genes previously shown to be up- (red) or down-regulated (blue) in a population of PROCR+ mammary stem cells show correlation with ZEB1+ cells in scRNAseq. **c** Immunofluorescence analysis of TCF4 protein expression (red) in combination with basal marker SMA (green) and DNA stain using DAPI (blue) within tissue sections from primary human reduction mammoplasty samples revealed that TCF4 is expressed in a subpopulation of basal (SMA+) cells. Scale bar = 25 μm. **d** Violin plot for expression of KRT14 by cell state showing highest expression in the myoepithelial (Myo) cells. **e** KRT14 and KRT8 double immunostaining revealed highest expression of KRT14 in ductal basal cells, while lobular basal cells show more diverse KRT14 positivity. Scale bar = 75 μm. See Supplementary Fig. 4 for violin plots displaying selected myoepithelial gene expression and identification of KRT8/KRT14 double positive cells

**Reconstructing lineage hierarchies within the epithelium**. To understand how these observed cell types and states are related to each other, we next reconstructed differentiation trajectories by pseudotemporal ordering of single cells using Monocle, which utilizes reverse graph embedding to generate a trajectory plot that can account for both branched and linear differentiation processes[33]. Applying Monocle to our droplet-based scRNAseq dataset on a subsampled population (4000 cells; 1000 cells per individual) from all four individuals yielded one tightly connected differentiation trajectory that separates into three main branches corresponding to the main cell types Basal, L1 and L2 (Fig. 6a). This suggests that the system is maintained through one continuous rather than several disconnected lineages. Considering the substantial evidence supporting the existence of MaSCs within the basal cell compartment[5,6], we manually set the start of pseudotime within the basal cell type

(Fig. 6b), thus resulting in a trajectory that differentiates into three main branches that are each enriched for Myo, L1 and L2, respectively. Of note, L1.2 is markedly enriched at the branching point between L1 and L2, suggesting that it represents a luminal-restricted bi-potent progenitor. It also precedes L1.1 on the L1 branch, suggesting that L1.2 is a progenitor to L1.1. Interestingly, L1.1 displayed high *ELF5* and *KIT* expression, which have been previously reported as progenitor cell markers[7,8]. Our data instead suggests that L1.1 represents a second mature, differentiated luminal cell type rather than a luminal progenitor that is upstream of L2. These basic principles were also reflected in our pseudotemporal analysis of the microfluidics-enabled scRNAseq dataset, which projects a bifurcation into luminal and basal lineage emerging from one common population of ZEB1 + progenitor cells (Supplementary Fig. 6a b). These results are in line with previous models of

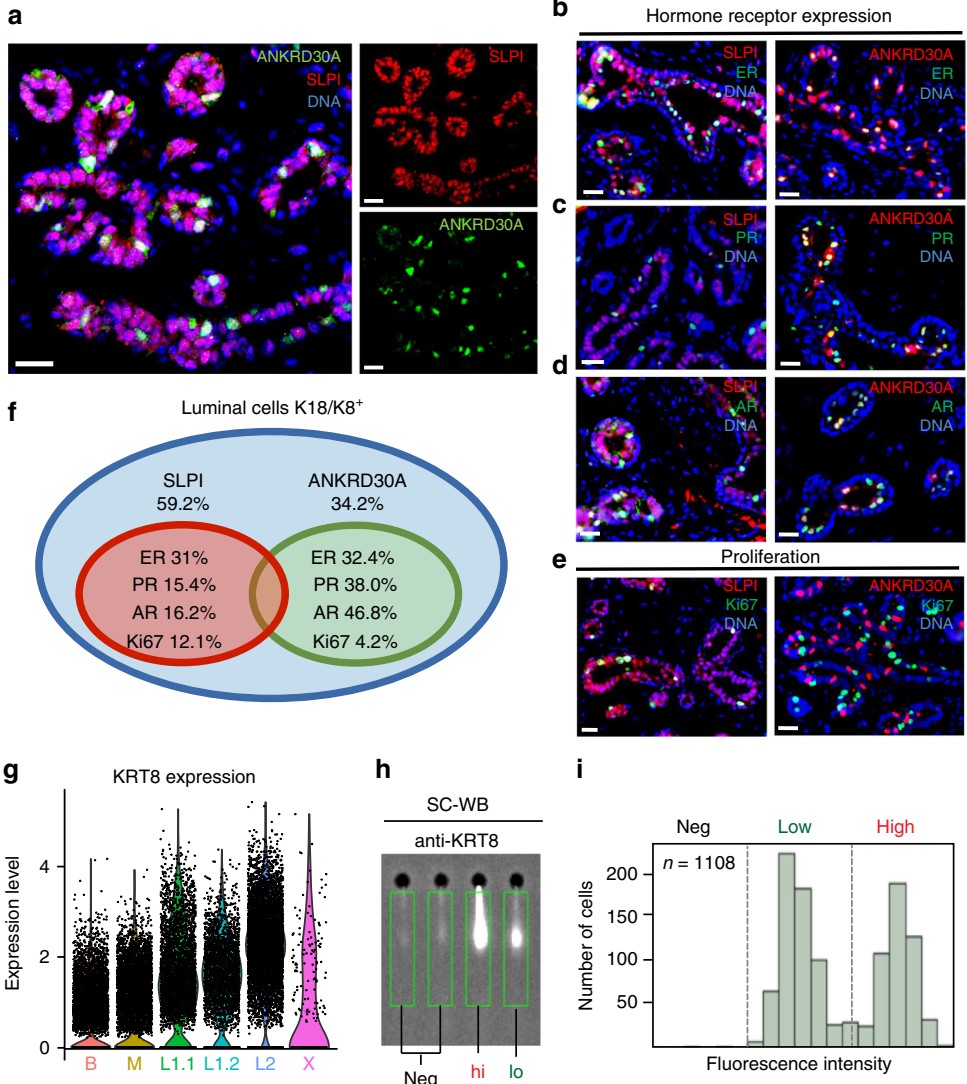

**Fig. 5** Validation and spatial integration of two distinct luminal cell types. **a** Immunofluorescence analysis of NY-BR-1 protein expression (green) in combination with basal marker SLPI (red) and DNA stain using DAPI (blue) within tissue sections from primary human reduction mammoplasty samples revealed that NY-BR-1 and SLPI are markers for distinct luminal subpopulations. **b–e** Immunofluorescence analysis of NY-BR-1 and SLPI (red) protein expression with: hormone receptors for estrogen receptor (**b**), progesterone (**c**), and androgen (**d**) and proliferation marker Ki67 **e** in green. **f** Summary of hormone receptor and proliferation marker expression in L1 and L2 cells. **g** Violin plot showing expression of KRT8 in the luminal subpopulations, higher expression is seen in the luminal L1.1 and L1.2 subpopulation. **h** Sample frame for detection of KRT8 protein content from individual cells using single cell Western blot following detection using microarray scanner. **i** Population summary showing cell number per fluorescence intensity confirmed bimodal distribution of KRT8 expression on the protein level. See Supplementary Fig. 5 for violin plots displaying expression of relevant hormone receptors as well as proliferation and luminal progenitor markers. All scale bars = 25 μm

mammary differentiation mediated by bi-potent stem/progenitor cells[4].

**Subpopulations correspond to breast cancer subtypes**. To learn more about the relationship of these newly defined subpopulations to existing subtypes of breast cancer, we used our gene scoring approach to directly compare the gene signatures of each population to gene signatures associated with each cancer subtype from the Metabric dataset[34]. This showed that both Luminal A and Luminal B subtypes of breast cancer are closely related to L2-type luminal cells (Supplementary Fig. 6c, top), which is in line with previous gene signature analyses of FACS-enriched basal, luminal progenitor, and mature luminal cells[9]. In addition, a recent report by Lehman et al. used global gene expression analyses to identify molecularly distinct subtypes within triple negative breast cancer (TNBC)[35]. We found that Myo showed

highest similarity to the mesenchymal-like subtype of TNBC, while the Basal1 class of TNBC yielded highest scores in the luminal L1.1 state (Supplementary Fig. 6c, bottom). Taken together, these analyses allow us to directly link several defined breast cancer subtypes to distinct cell populations of epithelial cells suggesting that the subtypes of breast cancer may arise from different tumor cells-of-origin.

## Discussion

The current state of knowledge in breast epithelial biology is largely based on population-level analyses of separated basal and luminal cells following bulk analyses of these distinct epithelial cell types[7]. While several distinct subpopulations of murine basal and luminal cells have been reported anecdotally[4], comprehensive knowledge about expression signatures and cellular identities of

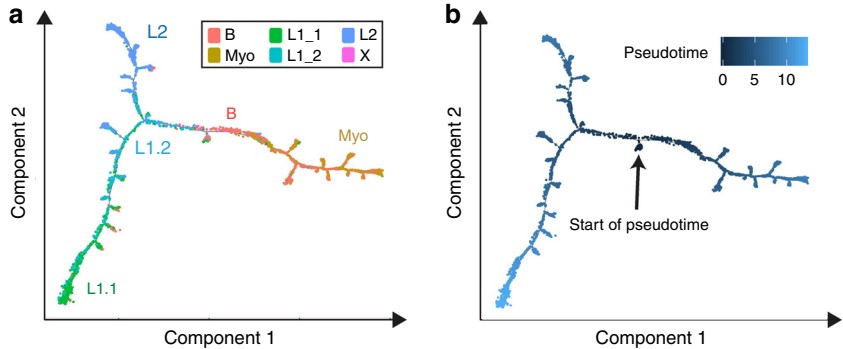

**Fig. 6** Reconstruction of differentiation and relation of cell states to breast cancer subtypes. **a** Monocle-generated pseudotemporal trajectory of a subsampled population of cells ($n = 4000$) from four individuals analyzed using droplet-mediated scRNAseq is shown colored by cell state designation. **b** Pseudotime is shown colored in a gradient from dark to light blue and start of pseudotime is indicated. See Supplementary Fig. 6 for summary list of discovered cell states, Monocle analysis of microfluidics-enabled scRNAseq results and gene scoring for breast cancer subtypes

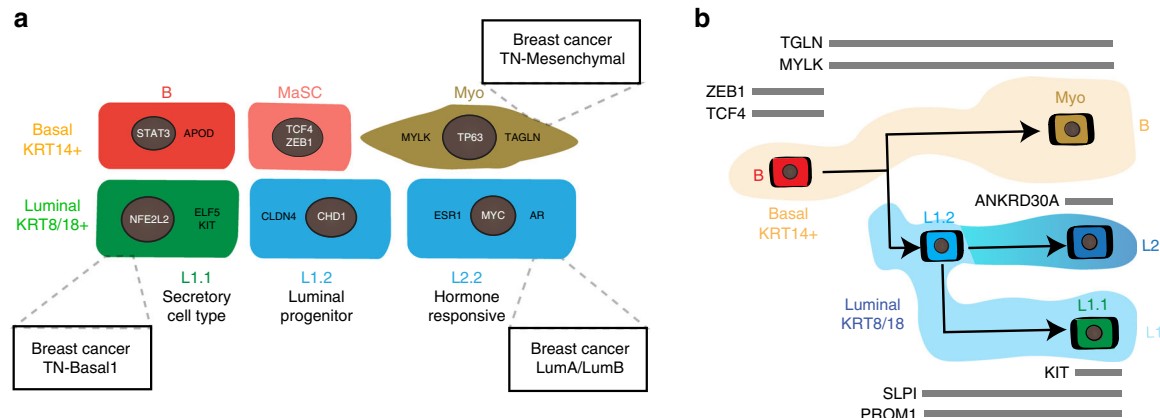

**Fig. 7** Proposed cellular heterogeneity and lineage hierarchies within the human breast. **a** Schematic summary of discovered cell states within the basal and luminal compartment of the human breast epithelium with proposed function, key transcription factors (in white), selected markers (in black) and similarities to breast cancer subtypes indicated in boxes. **b** Proposed model summarizing the lineage hierarchies within the breast epithelium based on one continuous differentiation trajectory from basal stem cells to three distinct differentiated cell types with overlaid marker genes of interest shown (black on gray bars)

these subpopulations remains sparse, particularly in the human system. Our scRNAseq analysis of the human breast epithelium from non-diseased, post-puberty, pre-menopause individuals for the first time allow for unbiased, de novo identification of distinct cell types and states in the adult human breast epithelium before pregnancy-induced changes occur. Strikingly, our approach revealed the existence of three main epithelial cell types (Basal, L1 and L2), in line with a recent scRNAseq analysis of the mouse mammary gland[36], although this work referred to these populations as "basal", "luminal progenitor" and "mature luminal cells". Our spatial analyses showed that these three cell types intermingle within ducts and lobules, and appear to form functionally distinct lineages that contribute to different aspects of breast biology (summarized in Fig. 7a). The fact that all three cell types contained a fraction of proliferative cells suggests that each cell type may be maintained by cycling, lineage-restricted progenitor cell subpopulations during normal homeostasis.

Our unbiased clustering analysis and pseudotemporal reconstruction of differentiation trajectories strongly suggest that these cell types represent three main branches of specified, differentiated cells, namely basal/myoepithelial, secretory L1, and hormone-responsive L2 cells (Fig. 7b). The lineage hierarchy likely starts with basal MaSCs[5,6] that differentiate either into specified myoepithelial cells, or into a common luminal progenitor, which gives rise to the two distinct luminal cell types L1

and L2. Interestingly, the *ELF5/KIT*-expressing subpopulation L1.1 represents a mature differentiated luminal cell state as it was predominantly located at the end of the L1 branch, suggesting that *ELF5/KIT* may be crucial for differentiation into the secretory L1 cell type, rather than promoting progenitor cell function as previously described[7,8]. It appears to be the L1.2 cell state within the L1 cell type that harbors a luminal-restricted bi-potent progenitor capacity for differentiation into the more specified secretory L1.1 or hormone-responsive L2 cells.

A currently unresolved question of active debate is whether MaSCs act as bi-potent stem cells that give rise to both lineages of basal and luminal cells[37], or whether homeostasis is mediated through distinct uni-potent, lineage-restricted basal and luminal stem cells[38]. Considering these two models, Monocle could have yielded a sparsely connected differentiation trajectory separating basal and luminal lineages, which would have supported a trajectory driven by lineage-restricted basal and luminal uni-potent progenitor cells on both ends of the spectrum. Instead, the outcome of our Monocle analysis is in favor of the existence of the bi-potent stem/progenitor model as it clearly identified one continuous trajectory indicative of a common source for both basal and luminal cell differentiation.

Understanding the origins of breast cancer in its earliest phases has the potential to advance methods of cancer early detection, and may ultimately form the basis to prevent cancer progression

before it turns into a life-threatening disease. Here, we asked whether the newly identified cell states correspond to specific subtypes of breast cancer, and thus may represent potential cancer cells-of-origin for the specific breast cancer subtypes. The luminal epithelial cell type L2 showed the clearest correlation with both Luminal A and B subtypes from the Metabric dataset[39], which is in line with previously reported similarities between a FACS-enriched population of mature luminal cells and the luminal-like breast cancer subtypes[9]. The fact that several L2 markers are independently known as breast cancer-associated antigens such as *SYTL2* and *ANKRD30A*[40], and that it shows highest expression of *CDKN1B*/p27 as a marker for potential breast cancer cells of origin[31] further corroborates the link between the hormone-responsive L2 cell type to breast cancer in general. Interestingly, the cell state closest related to the TNBC Basal subtype was found to be the luminal progenitor-like population L1.1. The concept that a luminal cell may be the cell-of-origin for basal-type breast cancer is not new and has been previously proposed in the context of *BRCA1*-driven disease[9]. Interestingly, those cell states containing subsets of proliferative cells, namely B, L1.1 and L2 are predominantly linked to breast cancer subtypes, which is line with previous reports showing an association of mammary epithelial cell proliferation in normal tissues with increased breast cancer risk[41].

In summary, our results provide crucial insights into the spectrum of cellular heterogeneity within the human breast epithelium in unprecedented resolution. Our unbiased analysis of the single-cell gene signatures from seven human individuals provide evidence for defined differentiation trajectories to maintain homeostasis in the adult human breast, as well as distinct subpopulations of both basal and luminal lineage that may serve as cells of origin for the different subtypes of breast cancer. Our single-cell atlas comprising the human breast epithelium will serve as a resource to map out the defined changes occurring during breast cancer and therefore form the basis for improved methods of cancer early detection and possibly strategies for cancer prevention.

## Methods

**Origin of tissue samples**. Anonymous reduction mammoplasty samples were acquired from NCI Cooperative Human Tissue Network (CHTN) and from Department of Surgery, Feinberg School of Medicine, Northwestern University. Other investigators may have received specimens from the same tissue specimens obtained through NCI CHTN. Specimens were anonymized then collected and distributed by CHTN, specimens are covered under collection/distribution of tissues under consent or waiver of consent. Samples were washed in PBS (Corning 21-031-CV) and mechanically dissociated using a razor blade. Dissociated samples were digested overnight in DMEM (Corning 10-013-CV) with Collagenase Type I, 2 mg/mL (Life Technologies 17100-017). Viable organoids were separated using differential centrifugation and viably frozen in 50% FBS (Omega Scientific FB-12), 40% DMEM, and 10% DMSO (Sigma-Aldrich D8418) by volume.

**Single-cell RNA sequencing**. Viable organoids were thawed and washed using DMEM, and digested with 0.05% trypsin (Corning 25-052-CI) containing DNase (Sigma Aldrich D4263-5VL) to generate single cell suspension. Cells were stained for FACS using fluorescently labeled antibodies for CD31 (eBiosciences 48-0319-42), CD45 (eBiosciences 48-9459-42), EpCAM (eBiosciences 50-9326-42), CD49f (eBiosciences 12-0495-82), and SytoxBlue (Life Technologies S34857). We only proceeded with samples showing at least 80% viability as measured using SytoxBlue in FACS.

Sorted cells were washed and resuspended at a concentration of ~500 cells/μl. For microfluidics-enabled scRNAseq, cell suspensions were mixed with Fluidigm C1 Suspension Reagents (Fluidigm 100-5315) at a ratio of 8:2 before loading mix onto C1 chip (Fluidigm 100-5760). Bright field images of captured cells were collected using a Keyence BZ-X710 microscope (Keyence Corporation, Itasca, Illinois, USA). Single-cell RNA isolation and amplification were performed using the Fluidigm C1 Single Cell Auto Prep IFC following the Fluidigm Protocol: 100-7168 I1. RNA spike-in controls were omitted. cDNA library preparation were performed following the Fluidigm C1 Protocol: 100-7168 I1.

For droplet-enabled scRNAseq, flow cytometry sorted cells were washed in PBS with 0.04% BSA and reseupended at a concentration of ~1000 cells/μl. Library

generation for 10× Genomics v1 chemistry was performed following the Chromium Single Cell 3′ Reagents Kits User Guide: CG00026 Rev B. Library generation for 10× Genomics v2 chemistry were performed following the Chromium Single Cell 3′ Reagents Kits v2 User Guide: CG00052 Rev B.

Quantification of cDNA libraries was performed using the Qubit dsDNA HS Assay Kit (Life Technologies Q32851) and high-sensitivity DNA chips (Agilent. 5067-4626). Quantification of library construction was performed using KAPA qPCR (Kapa Biosystems KK4824). For microfluidics-enabled scRNAseq libraries, we generally multiplexed 96 cells per lane on an Illumina HiSeq2500 resulting in a calculated depth of ~1.6 million reads per cell (Illumina Rapid PE kit v2 402-4002 and Rapid SBS kit v2 FC 401-4022). For droplet-enabled scRNAseq, we used the Illumina HiSeq4000 platform to achieve an average of 50,000 reads per cell.

**Processing of scRNAseq data**. After demultiplexing sequencing libraries to individual cell FASTQ files (observed average read depth per cell was found to be ~1.6 Million reads), each library was aligned to an indexed GRCh38 RefSeq genome using RSEM version 1.2.12[42], and bowtie2 version 2.2.3 with the following options enabled: rsem-calculate- expression -p $CORES—bowtie2—paired-end -output- genome-bam. Fragments Per Kilobase of transcript per Million mapped reads (FPKM) values were quantified and concatenated into a resulting gene expression matrix for each library, which was then loaded into R for subsequent computational analysis. For quality control filtering, we generally excluded libraries with less than 900 genes detected. In addition, genes that were not detected in at least 3 of the cells after this trimming were also removed from further analysis. Alignment of 3′ end counting libraries from droplet-enabled scRNAseq analyses was completed utilizing 10× Genomics Cell Ranger 1.3.1. Each library was aligned to an indexed GRCh38 genome using Cell Ranger Count. "Cell Ranger Aggr" function was used to normalize the number of confidently mapped reads per cells across the libraries from different individuals utilizing 10× v2 chemistry.

**Cluster identification using Seurat**. For cluster identification in both micro-fluidics- and droplet-enabled scRNAseq datasets, we utilized the Seurat pipeline[17]. The data matrices were imported into R and were processed with the Seurat R package version 1.2.1, where the FPKM values were transformed into log-space after the aforementioned trimming steps (each gene was expressed in at least three cells, each cell has at least 900 genes). PCA was performed using highly variable genes in the trimmed dataset. Using the first two PC's as input, we then performed density clustering to identify groupings in the data and t-distributed statistical neighbor embedding (tSNE) to visualize. Using further Seurat functionality, marker genes for each respective cluster were identified and used for subsequent analysis.

For droplet-enabled scRNAseq data, we used the Seurat R package version 2.0.0. Data was read into R as a counts matrix and transformed into log-space. Due to the difference in gene detection across the two platforms, differences in chemistry for the library prep, as well as sequencing depth per cell, a minimum cutoff of 500 and a maximum cut-off of 6000 genes per cell for this dataset was used. In addition, cells with a percentage of total reads that aligned to the mitochondrial genome (referred to as percent mito) greater than 10% were removed, since increased detection of mitochondrial genes can be associated with cells undergoing stress and cell death[43].

To account for the possibility of individual cell complexity driving cluster separation, we employed Seurat's "RegressOut" function to reduce the contribution of both the number of UMI's and the percent mito. Variable genes were then determined for subsequent PCA for each separate individual. For tSNE projection and clustering analysis, we used the first ten principal components. We used the feature plot function to highlight expression of known marker genes for basal (e.g., KRT5, KRT14) and luminal cells (e.g., KRT8, KRT18) to identify which clusters belonged to which epithelial cell type. The specific markers for each cluster identified by Seurat were determined using the "FindAllMarkers" function.

**Cluster comparisons and assignment**. Cluster specific marker genes from the individual library analyses were used as input lists to the previously described gene scoring method (described in more detail below) to compare cluster signatures in a pairwise manner between individuals. To visualize pairwise gene scoring results, we generated heatmaps displaying averaged gene scoring results for each cluster. We overlaid individual-specific cluster designations onto these heatmaps to find which individual clusters best match to each other. Clusters were merged together in the case that multiple clusters scored highly. We performed a separate Seurat analysis using combined basal cells from all four individuals, and then matched clusters using the gene scoring method on a set of genes curated to represent a myoe-pithelial cell fate[25] to score and classify the clusters as either Basal (B) or Myoe-pithelial (Myo) cell state.

**Gene scoring**. To compare gene signatures and pathways in epithelial sub-populations, we utilized individual gene scores as described previously[12]. Briefly, each score was generated by calculating total gene expression for each of the analyzed genes and separating them into 25 bins of similar expression. For every gene in each target pathway or signature, 100 "control" genes were selected from its corresponding bin and added to a "control" pathway. The resulting "control" pathway contained an equivalent expression distribution as the target pathway and

its average represents an equivalent sampling of 100 pathways of equal size to the target pathway. The expression of genes in the target pathway and the "control" pathways was averaged across each cell to generate a target score ($S_{Target}$) and control score ($S_{Ctrl}$). The cell's score for the target pathway ($S_{Path}$) is the difference between the target score and control score: $S_{Path} = S_{Target} - S_{Ctrl}$. To determine statistical significance, we used the unpaired Wilcox test with a 95% confidence interval.

**Gene set and pathway analysis**. Cells belonging to subpopulations were averaged to serve as a representation of each subgroup, and trimmed to their respective marker genes as determined by Seurat following log2 transformation. Each subpopulation sample was then uploaded to Ingenuity Pathway Analysis (Ingenuity Systems, www.ingenuity.com) core analysis feature and compared. A p-value of 0.05 was used as a cut-off to determine significant enrichment of a pathway or annotated gene grouping present in the Ingenuity Knowledge base. In addition, comprehensive gene set enrichment was done using Enrichr[26] based on the cell type and state specific marker genes identified by Seurat.

**Immunofluorescence analysis**. Tissues were fixed in 4% formaldehyde for 24 h, dehydrated in solutions of increasing concentrations of ethanol, cleared with xylene, and embedded in paraffin. Slides of 10-μm sections were prepared using a Leica SM2010 R Sliding Microtome (Leica Biosystems, Wetzlar, Germany). Slides were heated at 65 °C for 1 h, followed by two 5-min incubations in Histo-Clear (National Diagnostics, Cat. No. HS-200, Atlanta, Georgia, USA) for paraffin removal. Tissues were rehydrated with solutions of decreasing concentrations of ethanol, washed in double-distilled H$_2$O and PBS, and subjected to antigen retrieval using a microwave pressure cooker with 10 mM citric acid buffer (0.05% Tween 20, pH 6.0). Tissues were blocked in blocking solution (0.1% Tween 20 and 10% Goat Serum in PBS) for 20 min at room temperature, incubated with primary antibodies prepared in blocking solution at 4 °C overnight, washed in PBS, incubated with secondary antibodies diluted in PBS for 1 h at room temperature, and washed in PBS. Slides were mounted with VECTASHIELD Antifade Mounting Medium with DAPI (Vector Laboratories, Cat. No. H-1200, Burlingame, California, USA) and micrographs were taken with the BZ-X700 Keyence fluorescent microscope. For quantification of staining (e.g., ZEB1 and KRT14 staining), we manually counted positive cells as signal around nuclei (DAPI) and utilized the BZH Hybrid Cell Count software (Keyence) in at least three different fields of view using a 40× objective in at least two different samples.

Primary Antibodies: Estrogen Receptor (ER) rat mAb diluted 1:50 (Cat. No. 916201); KRT14 rabbit pAb diluted 1:500 (Cat. No. PRB-155P) (Biolegend, San Diego, CA, USA); Androgen Receptor (AR) rabbit mAb diluted 1:400 (Cat. No. 5153); Progesterone Receptor (PR) rabbit mAb diluted 1:1000 (Cat. No. 8757) (Cell Signaling, Danvers, MA, USA); KRT8 (TROMA-1) mouse mAb diluted 1:500 (DSHB, Iowa City, Iowa, USA); SLPI goat pAb diluted 1:200 (R&D Systems, Cat No. AF1274-SP, Minneapolis, MN, USA); α-Smooth Muscle Actin mouse mAb diluted 1:500 (Cat. No GTX60466), Ki67 mAb diluted 1:200 (Cat. No. GTX16667); TP63 rabbit pAb diluted 1:500 (Cat. No. GTX102425), MUC1 rabbit pAb diluted 1:500 (Cat. No. GTX15481), ACTA2 mouse mAb diluted 1:500 (Cat. No. GTX60466); TCF4 rabbit pAb diluted 1:500 (Cat. No. GTX54531); E-cadherin (DCH1) rabbit pAb diluted 1:500 (Cat. No. GTX100443); KRT18 rabbit pAb diluted 1:500 (Cat. No. GTX112978) (GeneTex, Inc., Irvine, California, USA); ACTA2 mouse mAb diluted 1:500 (Cat. No. MA511547); NY-BR-1 mouse mAb diluted 1:500 (Cat. No. MS-1932-P0); KRT14 mouse mAb diluted 1:100 (Cat. No. MA511599); and KRT18 mouse mAb diluted 1:100 (Cat. No. MA512104) (Thermo Fisher Scientific Inc., Carlsbad, California, USA).

Secondary Antibodies: Donkey anti-mouse Cy5.5-conjugated IgG (Novus Biologicals, Cat. No. NBP1-73774, Littleton, CO, USA); Goat anti-rabbit IgG conjugated with Alexa Fluor 568 and 488 (Cat. No. A21069 & A11034); Goat anti-mouse IgG conjugated with Alexa Fluor 568 and 488 (Cat. No. A11004 & A11001); Goat anti-rat IgG conjugated with Alexa Fluor 488 (Cat. No. A11006); Donkey anti-rabbit FITC-conjugated IgG (Cat. No. A16030); and Donkey anti-goat IgG conjugated to FITC and Alexa Fluor 568 (Cat. No. A16006 & A11057) (Thermo Fisher Scientific Inc., Carlsbad, California, USA).

**Single-cell western blot**. Single-cell western blots were completed using the Single-Cell Western instrument Milo, scWest chips, and reagents from ProteinSimple (San Jose, CA). A standard 6%T scWest chip was re-hydrated in 1× Suspension Buffer for 15 min at room temperature. A volume of 1 mL of flow cytometry-sorted human mammary epithelial cells (combined basal and luminal) at 100,000 cells/mL were settled in medium onto the scWest chip for 15 min at room temperature. Un-captured cells were washed away with 1 mL of media. Captured cells were lysed for 10 s, then individual cell protein lysates were electrophoretically separated for 1 min at 240 V, and proteins were UV-captured for 4 min. After running on Milo, the scWest chip was washed 2 × 10 min in 1× Wash Buffer, then probed for mouse anti-cytokeratin 8 (Abcam ab9023) at 200 μg/mL and rabbit anti-β-tubulin (Abcam ab6046) at 100 μg/mL for 2 h at room temperature. Primary antibodies were diluted in 1 × Wash Buffer (final) containing 5% (w/v) BSA. After 3 × 10-min washes in 1× Wash Buffer, the scWest chip was incubated with donkey anti-rabbit IgG Alexa 647 (A-31573 ThermoFisher

Waltham, MA) and donkey anti-mouse IgG Alexa 488 (A-21202 ThermoFisher) at 100 μg/mL in 1× Wash Buffer containing 5% BSA for 1 h in the dark at room temperature. The chip was then washed 3 × 15 min in 1× Wash Buffer, dried, and imaged using a Molecular Devices Genepix 4400A (Sunnyvale, CA) (Standard Blue Filter 500 gain, Standard Red Filter 600 gain). Images were saved as single-color tiffs and analyzed using Scout software (ProteinSimple).

**Reconstructing differentiation trajectories using Monocle**. Cell fate decisions and differentiation trajectories were reconstructed with the Monocle 2 package, which utilizes reverse graph embedding based on a user defined gene list to generate a pseudotime plot that can account for both branched and linear differentiation processes. For pseudotemporal analysis of breast epithelial cells in C1 data, we used Monocle version 2.2.0, ordered a combined set of cells from all three individuals on a list of marker genes as determined by Seurat analysis using up to 20 genes per cluster with least 0.5 power (Supplementary Data 2). Labels of basal and luminal cells respectively were assigned according to the identity of the cells from the initial cell sorting and ZEB1 positive cells were labeled based on expression level >0. For pseudotemporal analysis of droplet-based scRNAseq data, we first ordered the four individuals in Monocle 2.2.0 separately using cell type markers identified in the C1 analysis along with the top 20 marker genes for each subpopulation in Seurat. Next, for each of these four datasets, we identified genes differentially expressed between trajectory clusters (States), averaged the gene expressions values for all cells within each State, and generated a Pearson correlation matrix for these average gene expression value across States. We averaged the four correlation matrices into one matrix and kept only genes that had an average Pearson correlation of 0.8 with at least one other gene. Finally, we ordered a random subsample of 4000 cells (1000 cells from each individual) by the genes from our correlation analysis that overlapped with Seurat identified subpopulation marker genes (Supplementary Data 2).

**Comparison of subpopulations to breast cancer subtypes**. To learn more about the relationship of the newly defined normal breast epithelial subpopulations to the known breast cancer subtypes, we used the gene scoring method to compare each subpopulation to previously described triple negative breast cancer subtypes. To this end, we utilized the genes that are specifically up-regulated in each subtype as previously reported[35,39]. To compare each subpopulation to METABRIC derived molecular subtype signatures, the METABRIC microarray expression dataset was downloaded and processed using the R Bioconductor package Limma version 3.30.13. Samples were grouped by their annotated molecular subtype, and differentially expressed genes was calculated for each group. The top 20% of the upregulated genes as sorted by log-fold change were then used for downstream scoring.

**Code availability**. Custom scripts are available at: https://github.com/kessenbrocklab/Nguyen_Pervolarakis_Nat_Comm_2018.

**Data availability**. The authors declare that all data supporting the findings of this study are available within the article and its supplementary information files or from the corresponding author upon reasonable request. All RNAseq data quantified data matrices along with their associated meta data have been deposited in the GEO database under accession code GSE113197.

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

## Acknowledgements

We thank Ying Yu for technical assistance and animal handling. We are grateful to the Cooperative Human Tissue Network (CHTN) for providing the mammoplasty samples, and to Laura van't Veer and Denise Wolf for helpful advice on data analysis. This study was supported by funds from the National Cancer Institute (R01 CA057621 and U01 CA199315 to Z.W, R00 CA181490 to K.K., and K22 CA190511 to D.A.L.) and from the Chan/Zuckerberg Initiative (HCA-A-1704-01668 to K.K. and D.A.L.).

## Author contributions

K.K., D.A.L. and Z.W. designed research and supervised research; Q.H.N., D.M., A.T.P., E.W., R.K., E.J., D.A.L. and K.K. performed research; J.R., S.A.K. and A.G. contributed new reagents/analytic tools and biospecimens; N.P., K.B., R.T.D., I.D. and K.K. performed bioinformatic analyses; Q.H.N., N.P., D.A.L., Z.W. and K.K. wrote the paper manuscript, and all authors discussed the results and provided comments and feedback.

## Additional information

**Competing interests:** The authors declare no competing interests.

