## [Peer Review File · Nature Communications]

Reviewers' comments:

Reviewer #1 (Remarks to the Author):

The authors have almost completely redone the study and rewrote the manuscript. Importantly, they have significantly (almost 30x) increase the number of cells analyzed, which was one of the major weaknesses of the original submission. They have also redid all of the immunofluorescence studies and the images are now of much better quality. Overall the manuscript is significantly improved. However, it is a bit worrisome that several of the figures/data that were questioned by the reviewers were simply removed. In addition, several points still remain to be addressed:

1. The authors highlight a ZEB1/TCF4+ cell population as particularly interesting in Figure 1, but then in Fig 2 describing the 10x data they state that this population was lowly detected/uninterpretable.
2. The manuscript has several scientifically incorrect statements. For example "staining for ITF2, the gene product of TCF4" - first, the staining is for protein not gene and ITF2 is just another name for TCF4 - it would be better to avoid using different names for the same gene.
3. Figure 7 is not cited in the text.
4. Figure 7 implies that mesenchymal triple negative breast cancer may originate from myoepithelial cells, basal from the L1 fraction etc., which is a bit misleading as similarities in expression profiles does not necessarily mean that the tumor originated from that fraction. Especially for the myoepithelial cells such a suggestion is most likely not correct. Thus, it would be useful to revise this figure to avoid making such suggestions (just draw the relatedness different way that does not imply it's originated from that cell type).
5. The high expression of PGR in the basal and myoepithelial population (Suppl Fig 5) makes no sense as PR is a luminal marker and based on immunostaining, it has not been seen in other cell types. The authors need to validate this result.

6. In Figure 5, it would have been useful to perform multicolor immunofluorescence for the selected genes and known luminal/basal markers to conform the localization of the cells to the correct cell layer/type.

--

Reviewer #2 (Remarks to the Author):

This manuscript is considerably enhanced by the addition of substantial new data and an improvement of previous data. Most of my comments have been addressed and those that have not have been discussed to my satisfaction.

--

Reviewer #4 (Remarks to the Author):

My major concern for this study is the subpopulation detection.

1. This study contains many batches (e.g, 13 C1 chips in Fluidigm system, different individuals, different platforms (Fluidigm and droplet-seq)). Therefore a study, which compares results from batch-by-batch and mixed batches, is highly recommended to address the robustness and reproducibility of subpopulation detection results. For example, a cell is defined as L1 cell type when mixed with different individuals and batches. A separate analysis for this cell by using its own batch or different batch combinations (e.g., individuals, sequencing batches) are required to see whether this cell is still defined as L1.

2. During QC steps, they used minimal 500 genes and maximum 6000 genes per cell as cutoffs for droplet scRNA-seq, and 900 genes as a cutoff for Fluidigm C1 scRNA-seq. How do they choose

these cutoffs? Furthermore, the number of genes detected is also highly associated with cell types. For example, the average genes detected in iPS (human) cells is ~8000 (Fluidigm) while the average genes detected in GW16 is less than 5000 (Fluidigm):

<https://academic.oup.com/bioinformatics/article/32/16/2514/1743129#supplementary-data>

(Figure S6; Jiang, Peng, James A. Thomson, and Ron Stewart. "Quality control of single-cell RNA-seq by SinQC." *Bioinformatics* 32.16 (2016): 2514-2516.)

So filter out cells solely based on certain number of genes may result in losing certain cell types in downstream analysis.

Minor comments:

1. "scRNA-seq is completely unbiased" (page 5) is overstated. Both cell preparation and data analysis steps may induce bias. For example, different cell types could have different cell dissociation rate which results in biased cell population ratio estimation. Different scRNA-seq platform has its own bias. For example, the number of genes detected in droplet scRNA-seq is significantly lower than Fluidigm C1 system. There are many bias in scRNA-seq and we need to carefully control them.
2. Figure 1 (d): "correlation = 0.56". Is it coefficient of correlation (R or Rho)? What correlations they calculated – Pearson or Spearman or something else? Based on this figure, it does not look like the data is correlated but it still gets 0.56 coefficient. Is this due to a few dots (like the first and last dots) make a 'good' correlation? The outliers do affect correlations. A cross-validation or bootstrapping (random sample a fraction of dots and recalculate correlations, repeat it many times and calculate the median R or Rho) are recommended to check the robust of R or Rho.

SUMMARY OF REVISIONS

We thank the reviewers for their praise of our work and thoughtful comments and suggestions. We have now addressed all remaining concerns by including additional changes in our manuscript, changing Figure 7a, and by specifically responding to each concern in detail in our point-by-point response (see below). In addition, we made changes to the manuscript to comply with the editorial policies and we are uploading the requested checklist forms as Related Manuscript files. All changes to the manuscript are tracked highlighted in the word document.

We also added an additional section on “Data Availability” at the end of the Methods section of our manuscript. All data will be made accessible at Gene Expression Omnibus including raw .fastq files and quantified data matrices along with their associated meta data. The respective Accession numbers will be listed as soon as they are available.

POINT-BY-POINT RESPONSE TO REVIEWERS’ CONCERNS

Below we have summarized each reviewer’s concern in numbered lists. The authors’ responses are shown in blue.

Reviewer #1 (Remarks to the Author):

The authors have almost completely redone the study and rewrote the manuscript. Importantly, they have significantly (almost 30x) increase the number of cells analyzed, which was one of the major weaknesses of the original submission. They have also redid all of the immunofluorescence studies and the images are now of much better quality. Overall the manuscript is significantly improved. However, it is a bit worrisome that several of the figures/data that were questioned by the reviewers were simply removed.

Response by Authors

Several of the figures and data from the previous version of the manuscript were removed in this version due to addition of new single cell data sets. We needed to update the manuscript with figures that illustrate the droplet-based scRNAseq data along with revised comparison between the various data sets. This generated important new figures for the conclusions drawn.

In addition, several points still remain to be addressed:

1. The authors highlight a ZEB1/TCF4+ cell population as particularly interesting in Figure 1, but then in Fig 2 describing the 10x data they state that this population was lowly detected/uninterpretable.

Response by Authors

This is due to technical differences regarding the chemistry used in droplet- versus microfluidics-enabled scRNAseq platforms. The fact that these genes were not detected in our droplet-enabled scRNAseq dataset is most likely due to the lower gene detection capacity compared to the Smart-Seq2-based C1 approach used in Figure 1. The difference in gene detection sensitivity between the two pipelines is well established and has been systematically addressed in a recent study (Svensson et al, 2017, Nat Methods).

2. The manuscript has several scientifically incorrect statements. For example "staining for ITF2, the gene product of TCF4" - first, the staining is for protein not gene and ITF2 is just another name for TCF4 - it would be better to avoid using different names for the same gene.

Response by Authors

We apologize for the confusion. This has now been corrected and we only refer to TCF4 in this manuscript (Page 11).

3. Figure 7 is not cited in the text.

Response by Authors

Figure 7 was cited in the discussion section of the manuscript (Page 16).

4. Figure 7 implies that mesenchymal triple negative breast cancer may originate from myoepithelial cells, basal from the L1 fraction etc., which is a bit misleading as similarities in expression profiles does not necessarily mean that the tumor originated from that fraction. Especially for the myoepithelial cells such a suggestion is most likely not correct. Thus, it would be useful to revise this figure to avoid making such suggestions (just draw the relatedness different way that does not imply it's originated from that cell type).

Response by Authors

We fully agree and we have now changed the figure to illustrate relatedness rather than origination of the breast cancer subtypes by using dashed lines connecting the cell type to boxes. We also changed the figure legend to emphasize "similarity" rather than "connection" of these cells to breast cancer subtypes.

5. The high expression of PGR in the basal and myoepithelial population (Suppl Fig 5) makes no sense as PR is a luminal marker and based on immunostaining, it has not been seen in other cell types. The authors need to validate this result.

Response by Authors

We fully agree with Reviewer 1 in that PGR is exclusively found in luminal cells on the protein level. Our immunofluorescence analysis for PGR confirms that PGR is luminal-restricted.

However, we do detect transcripts within a subfraction of basal cells by scRNAseq. This is most likely due to discordance between gene expression and protein translation and additional layers of regulation of protein translation, which may only lead to actual detectable protein levels in luminal cells. We have now discussed this discordance in the manuscript to make this point clear (Page 12).

6. In Figure 5, it would have been useful to perform multicolor immunofluorescence for the selected genes and known luminal/basal markers to conform the localization of the cells to the correct cell layer/type.

Response by Authors

Unfortunately, we were not able to multiplex many of these markers due to overlap between the species of the primary antibodies in our studies. However, we have performed separate immunofluorescence experiments and have confirmed with confidence that NY-BR-1 and SLPI are exclusively in the luminal compartment as they were all positive for the luminal marker K18.

Reviewer #2 (Remarks to the Author):

This manuscript is considerably enhanced by the addition of substantial new data and an improvement of previous data. Most of my comments have been addressed and those that have not have been discussed to my satisfaction.

Response by Authors

Thank you!

Reviewer #4 (Remarks to the Author):

My major concern for this study is the subpopulation detection.

1. This study contains many batches (e.g. 13 C1 chips in Fluidigm system, different individuals, different platforms (Fluidigm and droplet-seq)). Therefore a study, which compares results from batch-by-batch and mixed batches, is highly recommended to address the robustness and reproducibility of subpopulation detection results. For example, a cell is defined as L1 cell type when mixed with different individuals and batches. A separate analysis for this cell by using its own batch or different batch combinations (e.g., individuals, sequencing batches) are required to see whether this cell is still defined as L1.

Response by Authors

We agree with Reviewer #4 that batch effects are a critical source of unwanted technical variation. The main issue in this regard is our microfluidic-enabled scRNAseq dataset, in which

many individual chips had to be combined to reach more powerful cell numbers for detection of subpopulations. In our analysis of the microfluidics-enabled scRNAseq dataset, we combined all cells from three individuals and from 13 chips in total. As shown in **Figure 1b**, we observed three populations (Basal, L1 and L2). Each of these populations was comprised of cells from all three individuals (Supplementary Fig. 1d), which rules out individual-specific effects. We can also rule out that these populations are the result of Chip-batches, since we used 13 chips in total and only three populations were detected, which means that these three populations had to come from multiple different chips. Most importantly, the biology underlying these three populations was confirmed using droplet-enabled scRNAseq of additional four individuals, where all cells were analyzed in one batch (as one sample) and yet highly comparable results emerged from this analysis showing three main populations with large overlap in their marker gene signatures to those detected using the microfluidics-enabled approach.

Incorporating the droplet-based sequencing data, due to technical differences between the libraries (i.e. different chemistries preparing the libraries, different sequencing depths etc), it is difficult to combine the cells into a single analysis. Instead we compared marker signatures characteristic of the observed subpopulations and assign cell type labels to cells in the larger analysis. From there, we used a batch-by-batch approach, and performed clustering on each individual's library. To do exhaustive comparisons of small groups of cells between each of the possible data set combinations to confirm the identity of cluster determination would result in an extremely high dimensional comparison that would be very difficult to interpret. Instead we worked on a by cluster basis within batch comparisons and found cluster designations that were most similar between the individuals based on the expression of marker genes and moved forward with these generalizable subpopulations.

2. During QC steps, they used minimal 500 genes and maximum 6000 genes per cell as cutoffs for droplet scRNA-seq, and 900 genes as a cutoff for Fluidigm C1 scRNA-seq. How do they choose these cutoffs? Furthermore, the number of genes detected is also highly associated with cell types. For example, the average genes detected in iPS (human) cells is ~8000 (Fluidigm) while the average genes detected in GW16 is less than 5000 (Fluidigm):

<https://academic.oup.com/bioinformatics/article/32/16/2514/1743129#supplementary-data> (Figure S6; Jiang, Peng, James A. Thomson, and Ron Stewart. "Quality control of single-cell RNA-seq by SinQC." *Bioinformatics* 32.16 (2016): 2514-2516.)

So filter out cells solely based on certain number of genes may result in losing certain cell types in downstream analysis.

Response by Authors

For our QC steps, trimming based on the number of genes expressed in an individual cell is our most relaxed QC filtering step for the reasons listed in the comment above and others. It is true that different cell types express different numbers of genes, and within our own data we see that in general the basal compartment cells have a lower average number of genes than their luminal counterparts. Our cutoffs were chosen in the hopes of eliminating low complexity cells (that in

turn could be indicative of low / poor amplification of cDNA or poor capture of transcripts from the original cell), so that the cells that we are confident that the cells we continue with allow us to make confident conclusions about the underlying biology that are not due to technical differences confounding our results. In our C1 analysis, we trim out a total of 136 cells across all libraries as compared to 351 due to other QC metrics (percent mitochondrial genes expressed, doublets / empty wells in microfluidics chip, etc) of the original 1144 cells sequenced. For our 10x analysis, we trimmed out a total of 181 cells due to our gene cutoffs out of the original 24646 cells sequenced. While it is true that these could be real cell types that we are losing, it is our belief that they are more likely poor quality cell libraries and as consequence are removed.

Minor comments:

1. “scRNA-seq is completely unbiased” (page 5) is overstated. Both cell preparation and data analysis steps may induce bias. For example, different cell types could have different cell dissociation rate which results in biased cell population ratio estimation. Different scRNA-seq platform has its own bias. For example, the number of genes detected in droplet scRNA-seq is significantly lower than Fluidigm C1 system. There are many bias in scRNA-seq and we need to carefully control them.

Response by Authors

We agree with this reviewer and changed the section on page 5 of the manuscript:

“This approach allows an unbiased analysis of the spectrum of heterogeneity within a population of cells”

There are indeed several technical biases associated with cell dissociation from their natural microenvironment, incorporation into droplets, etc. We were referring mostly to the scientifically open and discovery-based nature of these approaches, since no prior knowledge about previously reported genes of interest are being applied to understand the cellular complexity of the biological systems studied.

2. Figure 1 (d): “correlation = 0.56”. Is it coefficient of correlation (R or Rho)? What correlations they calculated – Pearson or Spearman or something else? Based on this figure, it does not look like the data is correlated but it still gets 0.56 coefficient. Is this due to a few dots (like the first and last dots) make a ‘good’ correlation? The outliers do affect correlations. A cross-validation or bootstrapping (random sample a fraction of dots and recalculate correlations, repeat it many times and calculate the median R or Rho) are recommended to check the robust of R or Rho.

Response by Authors

The coefficient stated is a Pearson correlation coefficient. The ‘high’ correlation comes from the very small number of total cells in the set that are ZEB1+, the majority of which are also TCF4+. We could constrain the analysis to only cells that have positive expression of either of these genes and calculate within the subpopulation but we feel that it would inflate the value.

REVIEWERS' COMMENTS:

Reviewer #4 (Remarks to the Author):

All of my concerns have been well addressed and this is a nice work indeed.